# IoT and UAV Integration in 5G Hybrid Terrestrial-Satellite Networks

**DOI:** 10.3390/s19173704

**Published:** 2019-08-26

**Authors:** Mario Marchese, Aya Moheddine, Fabio Patrone

**Affiliations:** SCNL Laboratory, Department of Electrical, Electronic, Telecommunications Engineering and Naval Architecture (DITEN), University of Genoa, 16126 Genoa, Italy

**Keywords:** IoT, 5G, UAV, satellite communications

## Abstract

The Fifth Generation of Mobile Communications (5G) will lead to the growth of use cases demanding higher capacity and a enhanced data rate, a lower latency, and a more flexible and scalable network able to offer better user Quality of Experience (QoE). The Internet of Things (IoT) is one of these use cases. It has been spreading in the recent past few years, and it covers a wider range of possible application scenarios, such as smart city, smart factory, and smart agriculture, among many others. However, the limitations of the terrestrial network hinder the deployment of IoT devices and services. Besides, the existence of a plethora of different solutions (short vs. long range, commercialized vs. standardized, etc.), each of them based on different communication protocols and, in some cases, on different access infrastructures, makes the integration among them and with the upcoming 5G infrastructure more difficult. This paper discusses the huge set of IoT solutions available or still under standardization that will need to be integrated in the 5G framework. UAVs and satellites will be proposed as possible solutions to ease this integration, overcoming the limitations of the terrestrial infrastructure, such as the limited covered areas and the densification of the number of IoT devices per square kilometer.

## 1. Introduction

The evolution of the telecommunication networks, which will lead the world to the Fifth Generation of Mobile Communications (5G), is characterized by a deep change in the telecommunication network infrastructure and in the employed technologies. 5G will have to support a higher number of users/devices requiring an Internet connectivity with different performance requirements and a higher number of applications and use cases. The Internet of Things (IoT) is one of these recently-emerging use cases, which will involve an even higher number of connected devices aimed at collecting and sending data with different purposes and over different application scenarios, such as smart city, smart factory, and smart agriculture.

In some cases, the terrestrial infrastructure is not enough to guarantee the typical 5G Key Performance Indicators (KPIs) due to its design and intrinsic limitations. Coverage is an example: the terrestrial infrastructure will never be able to cover certain areas, such as oceans and rural and remote ares where there is no chance of or economic advantage in building a terrestrial network such as the one in big cities. Resilience in case of temporary or permanent infrastructure unavailability is another key point. Flying technologies, such as communication satellites and Unmanned Aerial Vehicles (UAVs), can contribute to overcome the limitations of the terrestrial infrastructure, offering wider coverage, higher resilience and availability, and improving user’s Quality of Experience (QoE).

IoT can benefit from the UAVs and satellite integration in the 5G ecosystem in many ways, also beyond the coverage extension and the increase of the available bandwidth that these objects can offer. UAVs and, perhaps in the future, satellites may be an essential part of the service.

UAVs have been used in various sectors of our daily lives, allowing us to perform activities ranging from package delivery such as Prime Air from Amazon [1] to diving in water as done by the Boeing’s drone that can be transformed into a submarine [2]. For the IoT, they can be properly equipped to act as IoT gateways, collecting and forwarding data from IoT sensors located in the covered area to the authorized users connected to the Internet through the 5G network. Their position and their movement path can be arbitrarily set depending on how the IoT devices are distributed in the area to cover and on the amount and periodicity of the generated data, offering a customizable solution, also reducing the infrastructure cost. Besides, depending on the payload hardware, they can also act as intermediate nodes for multiple IoT solutions, offering a multi-technology IoT solution in a single node.

Communications satellites have been widely employed to extend the communications (telephony and Internet) throughout the world since almost fifty years ago. Currently, there are tens of satellite communication constellations with this purpose, and some others are planned to be deployed to offer global Internet connectivity through Low Earth Orbit (LEO) satellites [3,4]. Some preliminary studies already focused on the exploitation of satellite networks for the IoT, envisioning the data exchange between IoT devices and end users through satellite communications [5]. Satellites can act as intermediate nodes directly connecting to the IoT devices or through other intermediate collecting nodes (such as UAVs) in order to offer a dynamically-deployable real-time connectivity.

Within this context, this paper considers the impact of and need for the IoT environment, the role of UAVs and satellites, and the joint integration of these technologies in the future 5G.

The paper is structured as follows, Section 2 gives a detailed description about the main IoT solutions, both short and long range, commercial and standardized solutions, highlighting the differences among them and presenting their network infrastructure. Section 3 introduces use cases and related works about the UAV employment in the IoT, followed by the state-of-the-art about satellite employment in the IoT reported in Section 4. Section 5 lists the different possible solutions to integrate IoT, UAV, and satellites in the 5G paradigm, focusing on the solution we envision as the first step of the integration of the IoT in 5G hybrid terrestrial-satellite networks. Future challenges regarding this integration are stated in Section 6. Conclusions and future works are drawn in Section 7.

## 2. IoT Technologies

The IoT refers to the data exchange between a large amount of devices and sensors connected wirelessly without human interference. It can be used in various fields such as smart metering, smart buildings, and environmental monitoring. The number of IoT connected devices installed worldwide has increased exponentially from 15.41 billion in 2015 to 23.14 billion in 2018, and it is expected to reach 75.44 billion in 2025 [6].

IoT wireless technologies can be divided into two categories: short and long range. Table 1 contains a summary of short-range solutions and of long-range brand-defined approaches available on the market and also identified as commercial solutions in the following.

Long-range commercial solutions shown in Table 1 share the common network architecture in Figure 1. IoT devices transmit their data to a brand-defined gateway through a technology acting at the data link and physical layers where IoT application data are directly encapsulated. All involved protocols are strictly brand defined. Each gateway is a protocol converter encapsulating the private IoT application into an Internet protocol stack to be conveyed to the IoT cloud where a brand-defined server will apply an application layer conversion from the IoT application to an Internet application protocol (such as HTTP) suitable for end users.

In parallel with private commercial investments, plenty of standardization efforts in the IoT environment are promoted by various standardization bodies and special interest groups such as the Institute of Electrical and Electronics Engineers (IEEE), the Third-Generation Partnership Project (3GPP), WEIGHTLESS-SIG, the European Telecommunications Standards Institute (ETSI), and the DASH7 Alliance. All the proposed solutions fall in the category of the so-called Low Power Wide Area protocols (LPWA) [7]. Table 2 summarizes the different technical specifications of the solutions created by the aforementioned groups. The architecture of these technologies is similar to the one presented in Figure 1, but the brand-defined gateway is substituted by a base station of the cellular network.

Ensuring effective communication between different objects and guaranteeing a continuous bond among them is a critical goal for IoT. The application layer relying on the solutions listed in Table 2 offers a set of protocols for end-to-end message exchange [10]. All of them support the publish-subscribe model, which is a messaging pattern where message senders (publishers) categorize messages into classes without the receiver’s (subscriber) knowledge, and in turn, the subscriber shows interest in one or more message classes to receive their messages.
CoAP: Constrained Application Protocol for IoT applications, created by the IETF Constrained Restful Environment [11,12].XMPP: eXtensible Messaging and Presence Protocol, IETF Instant Messaging IM standard.MQTT: Message Queue Telemetry Transport linking embedded devices and networks with applications and middleware.AMQP: Advanced Message Queuing Protocol.DDS: Data Distributed Service, developed by the Object Management Group [13,14].

## 3. UAV and IoT

Unmanned Aerial Vehicles (UAVs) have witnessed an exceptional growth in the market in addition to the huge demand in the IoT area. Installing a UAV or collections of UAVs known as swarms for IoT industrial services is becoming a reality and a necessity for applications concerning public safety, pollution mitigation, etc. [15,16]. During the upcoming years, UAVs, often called drones, will be used in various fields such as homeland security [17], security and border surveillance [18], and goods transportation [19,20].

### 3.1. UAV Use Cases

In this section, we introduce some of the use cases where UAVs are integrated with IoT.
Military: UAVs have been used in the military field for over a decade. When a manned flight is too risky to be done, the unmanned ones can be used to provide the troops with a 24-h eye over a specific area. Besides, UAVs are used as weapons and equipment providers in case of a siege and are considered as great tools in the search operations for any lost or injured soldier. Now, UAVs are used by several armies around the world such as the U.S., British, Australian, and Norwegian armies [21]. Moreover, in 2015, the Defense Advanced Research Project’s Agency (DARPA) revealed the development of the “system-to-systems” approach that uses many low-cost drones forming a swarm whose aim is to destroy the enemies’ targets [22]. The U.S. Department of Defense, in 2016, launched 103 drones forming a swarm, which was considered as the world’s largest micro-drone swarm. It was capable of making decisions and self-healing actions [23].Earthquake and environmental disasters: UAVs equipped with IoT sensors can be instructed to fly over a damaged area, recording the necessary information such as videos, pictures, gas levels, etc. This collected information will help the rescue teams, allowing them to be adequately equipped when visiting this area. These drones can be also be used to deliver food, water, and medicaments to affected people [24]. UAVs were used in Haiti in 2013 for food and supply delivery [25,26] and in Nepal in 2015 to study the risk reduction of the disaster [27]. The work in [28] considered that such UAVs can act as hot spots or base stations to collect messages from people, to be then sent to their families.Disaster management: As stated before, UAVs can be used effectively in case of disaster. They can be used also in the management phase. For example, during the last great earthquake in Japan, UAVs were used to capture images of the damaged reactors at Fukushima Daiichi nuclear power plant and to assess the reconstruction efforts taking place in Fukushima prefecture [29].Crowd surveillance: UAVs can be used as a “security agent” at huge events such as musical concerts or sports tournaments, if provided with the suitable IoT devices, allowing them to monitor public spaces. Such drones can detect any abnormal situations by capturing pictures and videos and then alerting security agents to interfere physically, thus ensuring safety and security [24]. The work in [30], for example, used UAVs with IoT devices in crowd surveillance based on face recognition by capturing and processing videos.Real-time road traffic monitoring: Flying over a specific location UAVs, can collect real-time information about roads’ conditions and send them to a central server to be analyzed. Relevant information may be also sent to vehicle users to assist them about which routes to take [24].Meteorology: Instead of using specialized UAVs, any flying UAV having specific IoT sensors can be useful in collecting information in a specific area such as wind speed, humidity, and temperature, to be transmitted to a centralized server for weather forecasting at reduced costs [24].

### 3.2. UAV and IoT: State-of-the-Art

Studies have been carried out to show how UAVs can be integrated into the IoT world. The mentioned face recognition sensor crowd surveillance application [30] exploits a platform composed of UAVs and IoT devices. UAVs equipped with video cameras are connected to the ground station by using an LTE cellular network. When a security guard, for example, notices suspicious behavior, he/she commands the UAV to take a video of the involved people and applies face recognition to verify if someone has a criminal record. The work in [31] found an optimal flying path for UAVs equipped with IoT sensors by using a location-aware multi-layer information map and utility functions based on sensor density, flight time, energy, and risk. Genetic algorithms were used to maximize these utility functions. An evacuation support system based on IoT and UAVs was proposed in [32]: the system, consisting of IoT devices, is controlled by an intelligent agent (agent-based IoT) and autonomously determines a suitable plan to support a quick evacuation. The work in [33] created an Intelligent Home Security System (IHSS), which has the ability to monitor sensors and control actuators such as UAVs. Experimental results showed the effectiveness and usefulness of this IHSS system. IoT and UAVs are promising in the agriculture field: IoT devices would help monitor and control crop parameters and increase the quality and quantity of food. For that purpose, the work in [34] distributed heterogeneous IoT devices in a crop field to monitor and record environmental parameters.

## 4. IoT and Satellite

In most of the papers in the state-of-the-art about satellite networks’ employment for IoT applications, IoT devices are installed in rural environments where no terrestrial communication is available such as deserts, mountains, oceans, etc. [35], where the only communication means is by satellite. In other environments, a satellite/IoT system may provide a cost-efficient solution compared to other terrestrial technologies. The work in [5] provided an overview of a possible satellite/IoT system architecture based on an LEO satellite constellation, which allows obtaining lower propagation delay and lower information loss than GEO satellites. The work in [36] focused on a technique used to collect data called the Satellite-Routed Sensor System (SRSS). In this technique, the satellite is the entity responsible for data collection from distributed IoT sensors and for data transmission to ground stations. A tsunami detection system was introduced in [37]. The aim of this system was to help in the early prediction of any possible tsunami. For that large number of sensor terminals consisting of buoys and sensors, the satellite is considered as an intermediate node that forwards the gathered data from sensor terminals in remote areas to base stations where the analysis operations take place. The wide coverage of the satellite allows the collection of data from all the sensor terminals deployed all over Japan.

However, there are still many open challenges that require being solved before proposing a solution able to allow IoT communications properly through satellite networks, such as the choice of the best suitable IoT communication protocol, as pointed out in [38]. The authors in [39] investigated and proposed different models to reduce the impact of control and data message overhead when the IoT was supported over an integrated satellite-terrestrial network based on Information-Centric Networking (ICN). The employment of wireless and satellite systems to support M2M/IoT services in smart grid use cases was presented in [40]. The authors investigated the pros and cons of the possible exploitation of different wireless technologies, including satellite communications, to support the smart grid use case and, in particular, the support of M2M and Supervisory Control And Data Acquisition (SCADA) over satellite links. The work in [41] was a study of the use of the CoAP protocol over a random access satellite medium and aimed to test the reliability of the considered system and improve the congestion control. A similar study that focused on the configuration of both CoAP and MQTT in an integrated M2M satellite network was presented in [42].

Energy consumption is another challenge under investigation to fulfill the energy efficiency of 5G KPI also in satellite networks considering IoT/M2M data traffic flows [43].

## 5. IoT, UAVs, and Satellites in the Emerging 5G Technology

### 5.1. IoT and 5G

Considering the foreseen 5G framework, the International Telecommunication Union (ITU) grouped different applications with the same needs in terms of performance requirements and offered user QoE in three different use cases [44]. One of them is the massive Machine-Type Communication (mMTC), which is characterized by a very large number of connected devices located in small areas and typically transmitting a low volume of non-delay-sensitive data. Most IoT applications fall into this category. However, the plethora of IoT application scenarios is huge and poses different challenges, which in some cases, cannot be solved by simply enhancing the terrestrial infrastructure. For example, in some rural and remote areas, both the CAPEX and OPEX costs of building (or enhancing) and managing a terrestrial communication infrastructure are not justified by the economic returns. Solutions to extend the Internet connectivity to smart farming and smart villages, among other 5G application scenarios [45], need to go beyond the terrestrial infrastructure.

### 5.2. UAV/Satellite and 5G

The 3GPP carried out studies aimed at defining the possible role of Non-Terrestrial Networks (NTNs) in the 5G ecosystem in Release 15 [46] and is proposing solutions from the networking viewpoint to allow the integration between non-terrestrial and terrestrial networks in Release 16 [47]. The main difference among these solutions is related to which functionalities are implemented on-board non-terrestrial objects. They can act as relay nodes between 5G User Equipment (UE) and the 5G access point (5G-gNB) aimed at extending the 5G Radio Access Network (RAN) coverage or as a backbone support. Besides, an additional study is investigating the possible employment of Satellite Communication (SatCom) networks as active nodes in the 5G access operations, i.e., to employ satellites able to generate 5G cells acting as 5G gNBs [48]. Each of these solutions has its pros and cons, and it is not clear yet which could be the best one for the IoT use case.

Several research projects are ongoing to investigate the employment of satellites and UAVs in 5G and develop and test solutions to be implemented in the future within the 5G network. The work in [49] presented the results of the ongoing SAT5G project funded by the European Commission [50] about satellite communications positioning in 5G scenarios for enhanced Mobile Broadband (eMBB). The aim is to provide cost-effective plug and play SatCom solutions and to accelerate the 5G deployment in all geographies. They selected four different satellite use cases for eMBB, and their efforts focused on (1) offering efficient multicast/broadcast content delivery to network edges, (2) providing 5G connectivity to premises to complement the terrestrial ones, (3) providing 5G connectivity to moving platforms such as airplanes, and (4) providing 5G as a fixed backhaul in areas where it is difficult to deploy terrestrial connections such as lakes, islands, etc. The work in [51] introduced a 5G-oriented network architecture based on satellite communication and Mobile Edge Computing (MEC) to aid eMBB applications and guarantee the QoE for live streaming applications. The evaluation was done by using a real satellite link. Another project called SATis5presented a demonstrator for the integration of satellite and terrestrial 5G networks [52]. A recently kicked-off EU project addresses the integration of UAVs into 5G, 5G!Drones [53]. Its ultimate aim is to design, implement, and run various examinations of UAV use cases on top of the 5G infrastructure and to validate the 5G KPIs. The use cases to be considered are UAV traffic management, public safety/savings lives, situation awareness, and connectivity during crowded events.

### 5.3. IoT, UAV/Satellite, and 5G Integration

The integration between UAVs/satellites and the 5G terrestrial network with the aim to support the IoT use case can be achieved in different ways and leads to manifold benefits depending, as said above, on the role of non-terrestrial objects. UAVs and satellites can be employed in 5G application scenarios in case there is no other communication infrastructure, with the aim to collect and forward the generated data by IoT devices. UAVs can help extend the coverage of commercial IoT networks, previously mentioned in Section 2, by acting as mobile access points whose position and movement path can be properly set and dynamically adjusted. A possible first step for the integration of the aforementioned technologies is discussed in the following. Some preliminaries before presenting the different architectures are given:IoT devices may be equipped to be 5G UE. The drawback is that IoT devices are in most cases very simple objects whose size, hardware, and available resource constraints are very strict. Hosting 5G UE functionalities may be an excessive burden.UAVs can act as 5G UE, 5G-gNBs, or as transparent Relay Nodes (RNs). In the first option, UAVs communicate with the IoT devices through a wireless non-5G interface, e.g., one of the commercial or standardized LPWA solutions shown in Table 1 and Table 2. UAVs may be linked to a 5G cell generated by a satellite in visibility (5G-gNB on-board satellite) or get access to the 5G Core Network (CN) through a satellite acting as RN. In the second case, UAVs equipped to be 5G-gNBs allow the deployment of 5G cells on-demand and offer connectivity to devices equipped as 5G-UE. Concerning the third option, if the IoT devices are 5G-UE and the access to the 5G network takes place in the terrestrial segment, UAVs are “RNs transparent” to the 5G communication and only implement frequency conversion and radio frequency amplification to allow data transmission to the satellite.Satellites can act as 5G-gNBs if they are equipped with a regenerative payload or as RNs between 5G UE and 5G RAN if they are transparent payload satellites. The same considerations can be applied for the satellite gateways, which can act as 5G-gNBs or be linked to 5G-gNBs through terrestrial links.The terrestrial segment includes 5G CN and 5G gNBs (if they are not deployed elsewhere).

Figure 2, Figure 3, Figure 4, Figure 5, Figure 6 and Figure 7 show the possible network architecture configurations (from the user plane viewpoint), taking into account the 5G standardization progress for the non-terrestrial network reported in [47].

In detail, the choice of considering IoT devices as not 5G-oriented devices is depicted in Figure 2, Figure 3 and Figure 4. They are equipped with interfaces based on an IoT protocol stack such as the ones described in Section 2. In this case, the UAV acts as 5G-UE, taking charge of the conversion between IoT and the 5G protocol stacks. The difference among these architectures is in the role of the satellite, i.e., in the location of the 5G-gNB. In Figure 2, the satellite is a simple RN, an “extender”, whose only aim is to perform Radio Frequency processing and Frequency Switching (RF and FS) required to transmit from the UAV to the satellite link and vice versa. 5G-gNB is located on the ground close to the 5G-CN and the satellite link 5G-UE and 5G-gNB. In Figure 3, the satellite is still a simple RN, but 5G-gNB is located closer to 5G-UE. 5G-gNB is still located on the ground, and the satellite acts as a backhaul between 5G-gNB and 5G-CN. The main difference between architectures in Figure 2 and Figure 3 is that, in the latter case, 5G-gNB is located close to the edge of the network, easing the employment of technologies, such as MEC, and allowing a better interaction between 5G-UE and 5G-gNB due to the lower communication delay between them. In Figure 4, the 5G-gNB functionality is moved on-board the satellite, increasing its complexity and leading to additional challenges the 3GPP is studying [48]. Having a satellite-based flying 5G-gNB is a very interesting scientific and technical challenge involving problems and implementation issues to be tackled in both software and hardware. For the same motivations, it is not of immediate application. Satellites acting as RNs are the most realistic option for the near future. Satellites already deployed and implementing other “classical” satellite communication protocols (e.g., DVBS2) can be exploited for this purpose. Satellites equipped with 5G-gNB functionality need to be designed and launched for this purpose, which could take more time than the few years forecast for the 5G terrestrial network activation.

Figure 5, Figure 6 and Figure 7 picture the network architecture considering the IoT devices as objects equipped with all 5G components to let them directly get access to a 5G cell, i.e., IoT devices are 5G-UE. This solution is not compliant with all the currently available commercial IoT solutions and would impose a unique standard for IoT communications in 5G. The presence of a unique standard technology avoiding compatibility issues among different solutions would be a great advantage. Anyway, industries that have already gained access to the market and sold their products will not easily give up their market share. Moving 5G-UE to the IoT devices allows both UAVs and satellites to act as simple RNs, as in Figure 5, or to host 5G-gNBs on the satellite, as shown in Figure 6, implying the already mentioned increased burden in the hardware complexity and higher required resources, which need to be properly managed. The case of 5G-gNB on the UAV is shown in Figure 7. It introduces additional hardware/software complexity on-board the UAVs. Even if moving 5G-UE within IoT devices is a fascinating and scientifically stimulating solution, as the choice to have 5G-gNB on-board a satellite or a UAV was already discussed, from the practical viewpoint, these actions cannot be immediately implemented.

Table 3 summarizes the pros and cons of each identified solution.

The solutions we see as the possible first step of the integration between non-terrestrial and terrestrial networks within the 5G framework are the ones shown in Figure 2 and Figure 3.

## 6. Future Challenges

The aspects that strengthen the architecture in Figure 2 and Figure 3 open new challenges:Looking at the UAV protocol stack, UAVs could be equipped with one or multiple interfaces towards the IoT devices. In this way, they would act as collectors of data generated by devices based on a specific or multiple IoT technologies and protocols at the same time, contributing to the integration among the plethora of IoT solutions.UAVs could be equipped with an additional interface to allow them direct access to a radio base station 5G-gNB. In this way, when UAVs are located inside the coverage area of a terrestrial 5G cell, they can choose to send the collected data to the satellite or the radio base station. Data could be sent alternatively or simultaneously through these two links if useful, exploiting multi-path solutions (e.g., MP-TCP).Moving the 5G-gNB closer or farther to the 5G-UE may have controversial benefits whose impact is still to be investigated; for example, the communication delay, which is higher between 5G-UE and 5G-gNBs and lower between 5G-gNBs and 5G-CN in the architecture in Figure 2 and vice versa in the architecture in Figure 3 and the possible employment of MEC technology if the computational and storage capabilities are closer to the IoT devices.

The hardware and software complexity on-board the UAV would increase by considering these aspects, as well as the energy consumption, an issue for these flying objects, even if several technological solutions could relieve this drawback. For example, the complexity could be distributed among more UAVs flying in a swarm where all UAVs are linked together through short-range wireless interfaces (e.g., WiFi). One UAV is in charge of sending/receiving data to/from the satellites, and each of the other UAVs is equipped with one LPWA technology different from the others, to take care of a specific access technology.

Additional research challenges and technological issues need to be managed, concerning:Routing: 5G-UE comprises the user endpoints of 5G communications, i.e., the 5G CN is not able to see the multiple IoT devices linked to a UAV. A proper mechanism has to be implemented on-board UAVs to allow them to distribute data destined to IoT devices (e.g., commands, requests, etc.) correctly. The necessary solution may be similar to the one developed by the Network Address Translation (NAT) for IP. This problem is more challenging if more devices based on different IoT solutions are all linked to the same UAV through different interfaces because each solution uses its own format to address the devices, and in some cases, it is proprietary.Handover: UAVs may change their position by following a predefined path, and also, satellites do not always cover a fixed area if they are non-geostationary satellites. This situation implies that an IoT device cannot always be linked to the same UAV, and a UAV is not always linked to the satellite. Proper handover mechanisms are required to manage ongoing communications during these different kinds of handover events.Long-term storage: UAVs may be temporarily disconnected from the 5G CN due to predictable or unpredictable faults of the satellite links, both between UAVs and satellites or between satellites and satellite gateways. Communications can suffer from obstacles between UAVs and satellites, which increase the attenuation due to scattering. High fading attenuation affects the satellite communications especially at high-frequency bands: rain attenuation is not negligible in the Ka or higher frequency bands, and outage events may happen during heavy rain at the satellite gateway’s site. The duration of link disruptions can range from a few ms up to a few minutes depending on the outage reason (slow or fast fading, obstacle persistence, etc.). Allowing UAVs to store data on-board for medium/long periods may avoid wasting data, bandwidth, and contact opportunities for IoT devices. Of course, the increase of UAV hardware/software complexity introduced by this possible functionality needs to be considered.Security: Some IoT solutions (especially the commercial ones) have been designed, including data encryption mechanisms to guarantee secure data exchanges between IoT devices and users. However, further analysis should be carried out in order to evaluate the impact of UAVs and satellites (e.g., larger delays, possible outage and handover events, etc.) on the security and safety of the supported applications.

Figure 8 shows the architecture integrating IoT devices, UAVs, and satellites in the 5G paradigm along with the possible challenges previously mentioned.

## 7. Conclusions and Future Work

This paper describes the integration of IoT devices, UAVs, and satellites in the 5G environment. A brief description of the different IoT technologies was presented, along with the state-of-the-art, the different use cases of UAVs and IoT, and the role of satellites in the IoT. Different architectures were presented by differentiating the action of IoT devices, UAVs, and satellites. The approach where UAVs were 5G-UE, satellites were RNs, and 5G-gNB was located in the terrestrial portion was discussed as the first step for the mentioned integration. Possible challenges were also presented. Future works will concern the implementation of the proposed approach. Mathematical analysis and numerical simulations will be the next step of this research activity, which will involve the practical implementation of (at least) one of the proposed architectures through an emulated/simulated testbed.

## Figures and Tables

**Figure 1 sensors-19-03704-f001:**
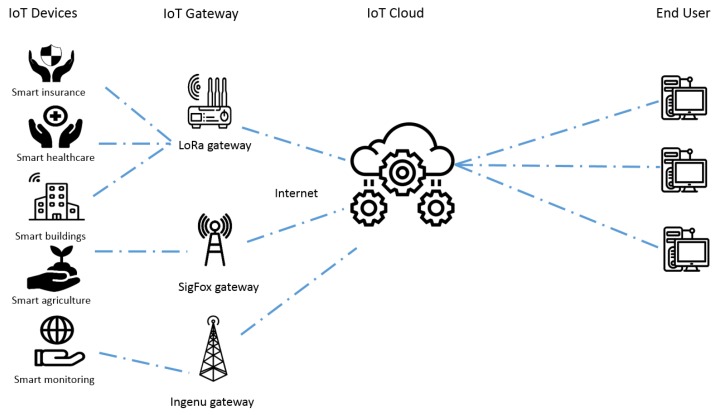
Long-range IoT commercial solutions architecture (icons are adopted from the Noun project website [9]).

**Figure 2 sensors-19-03704-f002:**
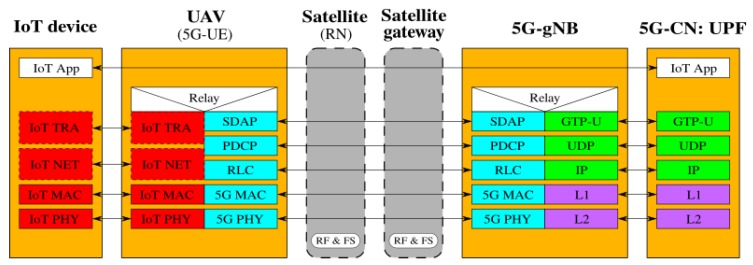
5G-UE on the UAV, satellite as RN, and terrestrial 5G-gNB.

**Figure 3 sensors-19-03704-f003:**
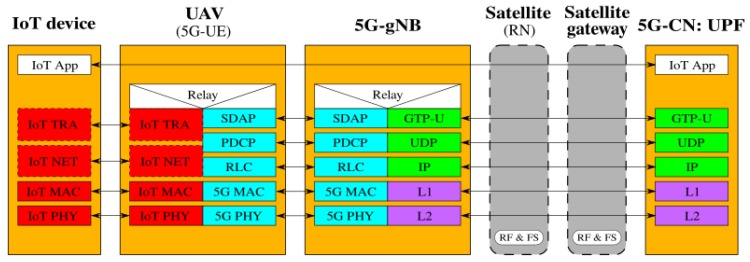
5G-UE on the UAV, 5G-gNB on the ground close to the UAV, and the satellite as the backhaul between 5G-gNB and 5G-CN.

**Figure 4 sensors-19-03704-f004:**
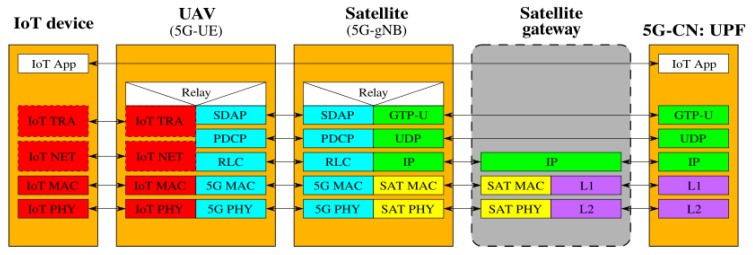
5G-UE on the UAV and 5G-gNB on the satellite.

**Figure 5 sensors-19-03704-f005:**
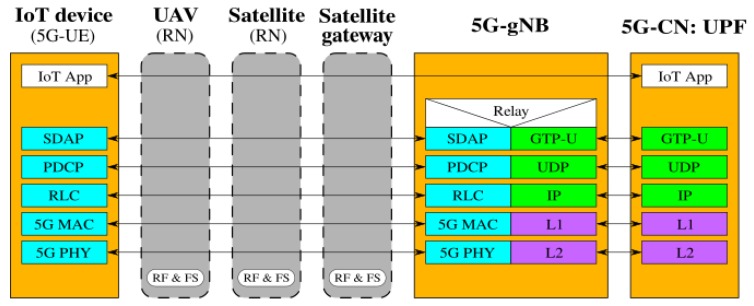
5G-UE in the IoT device, UAV and satellite as RNs, and terrestrial 5G-gNB.

**Figure 6 sensors-19-03704-f006:**
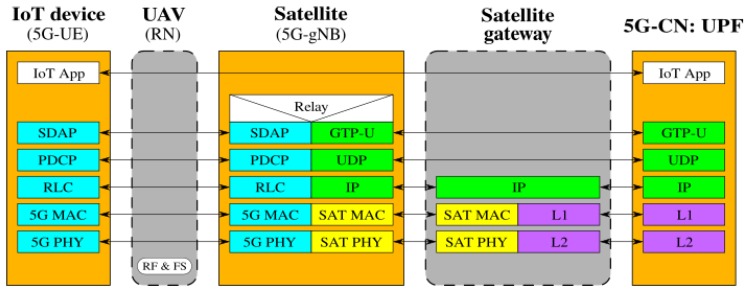
5G-UE in the IoT device and 5G-gNB on the satellite.

**Figure 7 sensors-19-03704-f007:**
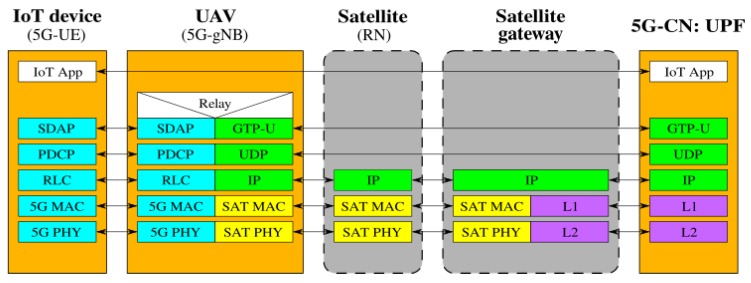
5G-UE in the IoT device and 5G-gNB on the UAV.

**Figure 8 sensors-19-03704-f008:**
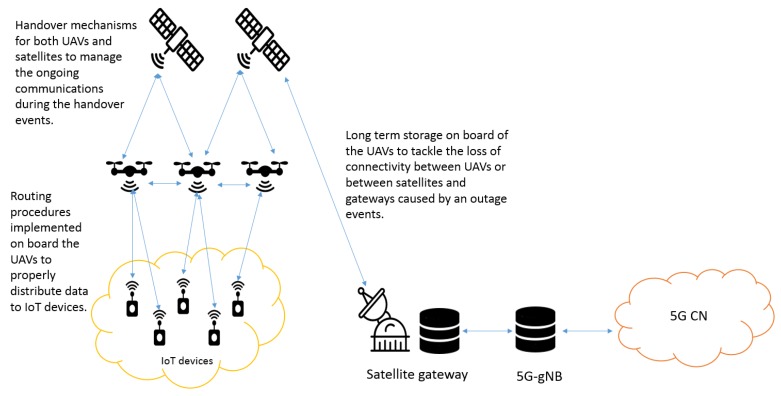
IoT device-UAV-satellite and 5G architecture.

**Table 1 sensors-19-03704-t001:** Comparison between short-range and long-range technologies [7,8].

Specifications	Short Range			Long Range		
	Bluetooth	ZigBee	WiFi	LoRaWAN	SigFox	Ingenu
Modulation	GFSK/DQPSK/DPSK	BPSK/OQPSK	various schemes	Chirp Spread Spectrum (CSS)	DBPSK(UL) GFSK(DL)	RPMA-DSSS(UL) CDMA(DL)
MAC	FDMA/TDMA	CSMA/CA	CSMA/CA	unslotted MAC	unslotted ALOHA	CDMA-like
Data rate	3 Mbps	250 kbps	7 Gbps	0.3 kbps–50 kbps	100 bps(UL) 600 bps(DL)	78 kbps(UL) 19 kbps(DL)
Coverage	up to 30 m	up to 100 m	up to 100 m	up to 5 km (urban) 15 km (rural)	10 km (urban) 50 km (rural)	up to 15 km (urban)

**Table 2 sensors-19-03704-t002:** Technical specifications for standardization bodies and special interest groups for IoT solutions [8]. ETSI, European Telecommunications Standards Institute.

Standardization Bodies and Special Interest Groups	Name	Modulation	Band	MAC	Data Rate	Coverage	Number of Channels
IEEE	802.15.4 k	DSSS, FSK	ISM SUB-GHz and 2.4 GHz	CSMA/CA, ALOHA with PCA	1.5–128 kbps	5 km (urban)	multiple (depends on channel and modulation)
	802.15.4 g	FSK, OFDMA, OQPSK	ISM SUB-GHz and 2.4 GHz	CSMA/CA	4.8–800 kbps	up to several km	multiple (depends on channel and modulation)
Weightless-SIG	-W	16QAM, DBPSK	TV white spaces (470–790 MHz)	TDMA/FDMA	1 kbps–10 Mbps	5 km (urban)	16 or 24
	-N	DBPSK	ISM SUB-GHz	ALOHA	30–100 kbps	up to 3 km (urban)	multiple, 200 Hz each
	-P	GPSK, QPSK	ISM SUB-GHz or licensed	TDMA/FDMA	200 bps–100 kbps	up to 2 km (urban)	multiple, 12.5 kHz each
DASHAlliance	DASH7	GFSK	SUB-GHz	CSMA/CA	9.6,55.6 or 166.7 kbps	up to 5 km (urban)	multiple, 25 or 200 kHz each
3GPP	EC-GSM	8PSK, GMSK	Licensed GSM	TDMA/FDMA	74–240 kbps	up to 15 km	124 channels, 200 kHz each
	NB-IoT	QPSK, 16QAM, 64QAM	Licensed LTE	SC-FDMA (UL) OFDMA (DL) 20 kbps (UL) 200 kbps (DL)	35 km	multiple, 180 kHz each	
	eMTC	QPSK, 16QAM, 64QAM	Licensed LTE	OFDMA/SC-FDMA	1 Mbps (UL,DL)	up to 15 km	multiple, 200 kHz each
ETSI	LTN	BPSK (UL) GFSK (DL) or OSSS	ISM SUB-GHz (433, 868 and 915 MHz)	BPSK (UL) GFSK (DL)	10–100 bps	up to 60 km	multiple, 200 Hz each

**Table 3 sensors-19-03704-t003:** Pros and cons of the identified integrated solutions.

	Role of					
Solution	IoT Device	UAV	Satellite	Sat.Gateway	Pros	Cons
Figure 2	not 5G oriented	5G-UE	RN	RN (or 5G-gNB)	simpler IoT devices and simpler bent-pipe 5G-agnostic satellites as the ones already available and deployed; 5G-gNBs’ complexity in the terrestrial segment’s nodes, which do not have strict resource constraints.	IoT devices not able to get direct access to the 5G network; more complex UAVs with high energy consumption due to protocol conversion.
Figure 3	not 5G oriented	5G-UE	RN	RN	simpler IoT devices and simpler bent-pipe 5G-agnostic satellites as the ones already available and deployed; 5G-gNBs’ complexity in the terrestrial segment’s, nodes which do not have strict resource constraints; better management of the 5G cell resources thanks to the 5G-gNB’s location closer to the edge of the network and the lower communication delay between 5G-UE and 5G-gNB.	IoT devices not able to get direct access to the 5G network; more complex UAVs with high energy consumption due to protocol conversion.
Figure 4	not 5G oriented	5G-UE	5G-gNB	RN	simpler IoT devices; regenerative satellites able to deploy 5G cells, increasing the reliability and availability of the terrestrial 5G access network.	IoT devices not able to get direct access to the 5G network; more complex UAVs with high energy consumption due to protocol conversion; more complex regenerative satellites with higher energy consumption, still to be designed, built, and launched.
Figure 5	5G-UE	RN	RN	RN (or 5G-gNB)	simpler and less resource-consuming UAVs and satellites, which can be based on non-5G technologies; 5G-gNBs’ complexity in the terrestrial segment’s nodes, which do not have strict resource constraints.	more complex and resource-consuming IoT devices equipped to get direct access to the 5G-RAN.
Figure 6	5G-UE	RN	5G-gNB	RN	simpler and less resource-consuming UAVs; regenerative satellites able to deploy 5G cells, increasing the reliability and availability of the terrestrial 5G access network.	more complex and resource-consuming IoT devices equipped to get direct access to the 5G-RAN; more complex regenerative satellites with higher energy consumption, still to be designed, built, and launched.
Figure 7	5G-UE	5G-gNB	RN	RN	UAVs able to deploy 5G cells on-demand for given locations and duration; simpler bent-pipe 5G-agnostic satellites as the ones already available and deployed.	more complex and resource-consuming IoT devices equipped to get direct access to the 5G-RAN; more complex and resource-consuming UAVs.

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
