# Peer review of "IoT and UAV Integration in 5G Hybrid Terrestrial-Satellite Networks"

_sensors, 2019, doi:10.3390/s19173704_

Round 1

Reviewer 1 Report

The manuscript is more like a tutorial and does not provide any sound scientific contribution.

I found all Sections except for Section 8.3 to provide already known information in a tutorial like fashion. Thus, these sections can be omitted.

Instead, the most interesting Section is Section 8.3 which describes various integration scenarios for Satellite, 5G, UAV and IoT. This is the main contribution of the paper.

Thus, the manuscript should be shortened and focus only on the progress beyond SoA. To this end, Section 8.3 needs to be further elaborated by providing pros and cons for each proposed architecture.

The manuscript also lacks any mathematical analysis as well as any validation or numerical simulation result.

Abstract should be rewritten as it does not contain anything about UAV although the title and the main text address the topic of UAV.

3GPP acronym is not defined correctly in the main text, but is defined correctly in the list of acronym.

References to relevant EU/ESA funded work addressing the integration of Satellite into 5G are missing, such as SaT5G (EC H2020 5GPPP), 5G-VINNI (EC H2020 5GPPP) and SATis5 (ESA ARTES) projects. Also, references to recently kicked-off EU projects addressing the integration of UAV into 5G are missing, such as 5G!Drones (EC H2020 5GPPP).

Author Response

We would like to thank the reviewer for the valuable suggestions.

The manuscript is more like a tutorial and does not provide any sound scientific contribution.

Actually, the paper has been submitted as a review paper about the current state-of-the-art of IoT technologies and their employment in the upcoming 5G system exploiting UAVs and satellite communication networks. The provided scientific contribution is limited to the description of the considered possible network architecture regarding IoT services in 5G hybrid terrestrial-UAV/satellite networks and the identified open challenges reported in Section 5.3 (previous Section 8.3).

Anyway, we have tried to increase the scientific novelty of the paper as indicated in the following.

I found all Sections except for Section 8.3 to provide already known information in a tutorial like fashion. Thus, these sections can be omitted. Instead, the most interesting Section is Section 8.3 which describes various integration scenarios for Satellite, 5G, UAV and IoT. This is the main contribution of the paper. Thus, the manuscript should be shortened and focus only on the progress beyond SoA.

Done. We have reduced the content of the previous Sections 2, 3, and 4 grouping them together in order to shorten the first part of the manuscript and better focus the contribution on Section 5.3 (previous Section 8.3).

To this end, Section 8.3 needs to be further elaborated by providing pros and cons for each proposed architecture.

Done. A brand new table (Table 3) has been added to report in a more concise way the role of each entity (IoT device, UAV, and satellite) and to summarize and better highlight the pros and cons of each identified solutions.

The manuscript also lacks any mathematical analysis as well as any validation or numerical simulation result.

As already mentioned, the paper has been submitted as a review paper. The novelty stands in the architectural proposal. Mathematical analysis and numerical simulations will be the next step of this research activity which will involve the practical implementation of (at least) one of the proposed architecture through an emulated/simulated testbed.

Abstract should be rewritten as it does not contain anything about UAV although the title and the main text address the topic of UAV.

Done. We totally rewrote the abstract in order to better summarize the investigated topics and the contents of the manuscript.

3GPP acronym is not defined correctly in the main text, but is defined correctly in the list of acronym.

We fixed this mistake and we ordered the acronyms in the Abbreviation list in alphabetical order in order to ease the search.

References to relevant EU/ESA funded work addressing the integration of Satellite into 5G are missing, such as SaT5G (EC H2020 5GPPP), 5G-VINNI (EC H2020 5GPPP) and SATis5 (ESA ARTES) projects. Also, references to recently kicked-off EU projects addressing the integration of UAV into 5G are missing, such as 5G!Drones (EC H2020 5GPPP).

We included in the references and added details about the mentioned research projects in Section 5.

Reviewer 2 Report

With the coming 5G wave, more advanced network connectivity with different performance requirements is needed, e.g., IoT, UAV, and satellite. Thus, this article considers and studies the impacts of IoT, UAV, and satellite in 5G systems, and the joint integration of these technologies is also analyzed. To be specific, the detailed description about IoT technique is first introduced and then some current solutions to integrate IoT, UAV and satellite in the 5G paradigm are given. Additionally, the authors state the future challenges regarding the integration of these techniques.

In general, the idea of this overview article is valuable, but I still have some comments as follows:

1. Section 2 and 3 are short and Section 2, 3, and 4 can be combined into one section. In addtion, three IoT techniques on short range in Section 3 and long range in Section 4 can be compared in the form of Table, respectively, which is easy to understand and read for the readers.

2. Compared with IoT and UAV, the introduction about satellite seems to be less, so more introduction and references about satellite can be added.

3. Note that there are many overview articles in the references, in my opinion, some popular articles combining these technologies can be added for showing the current research trends.

4. The abstract doesn't summarize the main idea of this article well, e.g., there is only IoT introduced in the abstract, so the authors should rewrite the abstract.

5. The whole article consists of some simple descriptions, lacking the simulation results. Thus, several simulation results can be used to verify the effectiveness of the integration of these technologies.

Author Response

We would like to thank the reviewer for the valuable suggestions.

With the coming 5G wave, more advanced network connectivity with different performance requirements is needed, e.g., IoT, UAV, and satellite. Thus, this article considers and studies the impacts of IoT, UAV, and satellite in 5G systems, and the joint integration of these technologies is also analyzed. To be specific, the detailed description about IoT technique is first introduced and then some current solutions to integrate IoT, UAV and satellite in the 5G paradigm are given. Additionally, the authors state the future challenges regarding the integration of these techniques.

In general, the idea of this overview article is valuable, but I still have some comments as follows:

We would like to thank the reviewer for his/her positive comment.

Section 2 and 3 are short and Section 2, 3, and 4 can be combined into one section. In addition, three IoT techniques on short range in Section 3 and long range in Section 4 can be compared in the form of Table, respectively, which is easy to understand and read for the readers.

Done. We merged the three sections and shortened the contents also by using a new table (Table 1) which reports a comparison between the mentioned short and long range technologies, previously described in the text.

Compared with IoT and UAV, the introduction about satellite seems to be less, so more introduction and references about satellite can be added.

Done. We extended the contribution on the state-of-the-art about IoT and satellite including four other recent papers on this topic.

Note that there are many overview articles in the references, in my opinion, some popular articles combining these technologies can be added for showing the current research trends.

We proceeded including a description of the currently ongoing research projects whose aim is to investigate the employment of satellites and UAVs in 5G and develop and test possible solutions to be implemented in the future within the 5G network. This contribution has been included in Section 5.

The abstract doesn't summarize the main idea of this article well, e.g., there is only IoT introduced in the abstract, so the authors should rewrite the abstract.

We totally rewrote the abstract in order to better summarize the investigated topic and the contents of the manuscript.

The whole article consists of some simple descriptions, lacking the simulation results. Thus, several simulation results can be used to verify the effectiveness of the integration of these technologies.

Actually, the paper has been submitted as a review paper about the current state-of-the-art of IoT technologies and their employment in the upcoming 5G system exploiting UAVs and satellite communication networks. Mathematical analysis and numerical simulations will be the next step of this research activity which will involve the practical implementation of (at least) one of the proposed architecture through an emulated/simulated testbed and will be included in the following contributions.

Anyway, we have tried to increase the scientific novelty of the paper by focusing better on the architectural proposal in Section 5.3 and by adding a brand new table (Table 3) containing the pros and cons of each proposed architecture.

Round 2

Reviewer 1 Report

MAJOR COMMENT:

Section 5.3: It is stated that “The solution we see as the possible first step of the integration between non-terrestrial and terrestrial networks within the 5G framework is the one shown in Figure 2.” However, there is an additional even more straightforward architecture which has not been addressed by the authors. This is the following:

IoT Device (not 5G oriented) <--> UAV (5G UE) <--> 5G gNB <--> Satellite (RN) <--> Satellite Gateway <--> 5G CN

According to this architecture, the satellite network provides backhauling between the 5G gNB and the 5G CN, which is straightforward and constitutes to the possible first step of the integration between non-terrestrial and terrestrial networks within the 5G framework. This architecture option should be included in the revised manuscript and the authors should elaborate on its description.

OTHER COMMENTS:

Page 6 / Line 180: Typo. “differnet” à “different”.

Page 6 / 1st Paragraph: The following reference is relevant to the SoA review reported in this paragraph and is currently missing:

Soua, M. R. Palattella and T. Engel, "IoT Application Protocols Optimisation for Future Integrated M2M-Satellite Networks," 2018 Global Information Infrastructure and Networking Symposium (GIIS), Thessaloniki, Greece, 2018, pp. 1-5. doi: 10.1109/GIIS.2018.8635784

Page 6 / Line 216: Typo “ongoing SAT5G project funded by ESA” à “ongoing SAT5G project funded by the European Commission”.

Citation in Reference [49]: Typo: “Integrating SatCom into 5G” à “SaT5G - Satellite and Terrestrial Network for 5G”.

Section 5.3 / Table 3 / page 10: Pros and Cons should be reported in separate columns. That is, one column for Pros and separate column for Cons.

Section 7: The authors claim in their responses that “Mathematical analysis and numerical simulations will be the next step of this research activity which will involve the practical implementation of (at least) one of the proposed architecture through an emulated/simulated testbed.” However, this statement is currently missing from the relevant Section 7 “Conclusions and Future Work” and should be added in the revised manuscript.

Author Response

We would like to thank the reviewer for the valuable suggestions.

Section 5.3: It is stated that “The solution we see as the possible first step of the integration between non-terrestrial and terrestrial networks within the 5G framework is the one shown in Figure 2.” However, there is an additional even more straightforward architecture which has not been addressed by the authors. This is the following:

IoT Device (not 5G oriented) <--> UAV (5G UE) <--> 5G gNB <--> Satellite (RN) <--> Satellite Gateway <--> 5G CN

According to this architecture, the satellite network provides backhauling between the 5G gNB and the 5G CN, which is straightforward and constitutes to the possible first step of the integration between non-terrestrial and terrestrial networks within the 5G framework. This architecture option should be included in the revised manuscript and the authors should elaborate on its description.

The suggested architecture has been included and considered among the others in Section 5.3. A dedicated figure (Figure 3) and some considerations about this further possible architecture have been added, highlighting the differences with the other solutions, especially the other ones where the IoT devices are not 5G-oriented devices. Table 3 has been modified adding a row related to the suggested architecture. An additional future challenge has been added in Section 6 related to the suggested architecture.

Page 6 / Line 180: Typo. “differnet” à “different”.

Fixed.

Page 6 / 1st Paragraph: The following reference is relevant to the SoA review reported in this paragraph and is currently missing:

Soua, M. R. Palattella and T. Engel, "IoT Application Protocols Optimisation for Future Integrated M2M-Satellite Networks," 2018 Global Information Infrastructure and Networking Symposium (GIIS), Thessaloniki, Greece, 2018, pp. 1-5. doi: 10.1109/GIIS.2018.8635784

Done. The reference has been added ([42]).

Page 6 / Line 216: Typo “ongoing SAT5G project funded by ESA” à “ongoing SAT5G project funded by the European Commission”.

Fixed.

Citation in Reference [49]: Typo: “Integrating SatCom into 5G” à “SaT5G - Satellite and Terrestrial Network for 5G”.

Fixed.

Section 5.3 / Table 3 / page 10: Pros and Cons should be reported in separate columns. That is, one column for Pros and separate column for Cons.

Done. The last column in Table 3 has been split into two separate columns: the Pros one and the Cons one.

Section 7: The authors claim in their responses that “Mathematical analysis and numerical simulations will be the next step of this research activity which will involve the practical implementation of (at least) one of the proposed architecture through an emulated/simulated testbed.” However, this statement is currently missing from the relevant Section 7 “Conclusions and Future Work” and should be added in the revised manuscript.

Done. The suggested statement has been added in Section 7 to better highlight how we would like to proceed in the first step of our future work on this research topic.